# Fine-Grained Recognition of Mixed Signals with Geometry Coordinate Attention

**DOI:** 10.3390/s24144530

**Published:** 2024-07-13

**Authors:** Qingwu Yi, Qing Wang, Jianwu Zhang, Xiaoran Zheng, Zetao Lu

**Affiliations:** 1National Key Laboratory of Radar Signal Processing, Xidian University, Xi’an 710071, China; 13933827718@139.com; 2State Key Laboratory of Satellite Navigation System and Equipment Technology, The 54th Research Institute of CETC, Shijiazhuang 050081, China; mr_w7364@163.com; 3 School of Communication Engineering, Hangzhou Dianzi University, Hangzhou 310018, China; mc170805@163.com (X.Z.); 221080042@hdu.edu.cn (Z.L.)

**Keywords:** machine learning, fine-grained image recognition, residual neural network, information geometry

## Abstract

With the advancement of technology, signal modulation types are becoming increasingly diverse and complex. The phenomenon of signal time–frequency overlap during transmission poses significant challenges for the classification and recognition of mixed signals, including poor recognition capabilities and low generality. This paper presents a recognition model for the fine-grained analysis of mixed signal characteristics, proposing a Geometry Coordinate Attention mechanism and introducing a low-rank bilinear pooling module to more effectively extract signal features for classification. The model employs a residual neural network as its backbone architecture and utilizes the Geometry Coordinate Attention mechanism for time–frequency weighted analysis based on information geometry theory. This analysis targets multiple-scale features within the architecture, producing time–frequency weighted features of the signal. These weighted features are further analyzed through a low-rank bilinear pooling module, combined with the backbone features, to achieve fine-grained feature fusion. This results in a fused feature vector for mixed signal classification. Experiments were conducted on a simulated dataset comprising 39,600 mixed-signal time–frequency plots. The model was benchmarked against a baseline using a residual neural network. The experimental outcomes demonstrated an improvement of 9% in the exact match ratio and 5% in the Hamming score. These results indicate that the proposed model significantly enhances the recognition capability and generalizability of mixed signal classification.

## 1. Introduction

Automatic modulation recognition (AMR) technology is used to identify the modulation schemes of received signals and find extensive applications in both military and civilian domains, closely related to non-cooperative communication, electronic warfare, and security detection [1,2,3]. With the rapid development of wireless communication technologies, the modulation types of signals are becoming increasingly diverse and complex. Additionally, the radio frequency environment is becoming more challenging, leading to frequent occurrences of signal time–frequency overlap [4]. Currently, research on automatic recognition of single modulation types has become relatively mature. For instance, methods employing decision theory, likelihood approaches, classification models based on domain knowledge and manually designed features [5,6], and neural network models based on deep learning [7] have been developed. However, research on the recognition of signals involving two or more modulation types is less explored. There are two main approaches for modulation type recognition in the context of multiple signal mixtures: one involves preprocessing by separating signals first and then recognizing the modulation type of each isolated signal, and the other approach aims to directly recognize multiple signals without extensive preprocessing or separation.

In the first approach, the effectiveness of signal classification is closely tied to the performance of signal separation. Commonly used algorithms for multi-signal separation include Independent Component Analysis (ICA) and its variant, Fast Independent Component Analysis (FICA) [8,9,10]. These methods frequently necessitate substantial prior knowledge to separate overlapping signals using channel matrices, which may be restrictive in practical applications.

Yang et al. employed a Uniform Linear Array (ULA) as a signal receiver to capture and distinguish signals. They utilized advanced techniques such as high-order cumulants and instantaneous statistical amplitude features, combined with a classical decision tree, for digital signal modulation classification. This method yielded positive results, even under conditions of low signal-to-noise ratio [11]. However, the effectiveness of this approach heavily depends on the precise estimation of signal arrival angles, and the classification performance is closely tied to this accuracy. Moreover, the use of ULA for signal reception and separation is more appropriate for scenarios where the signal reception matrix is either positive definite or overdetermined, which limits its use in complex settings dominated by underdetermined reception matrices. Ai et al. implemented a UNet semantic segmentation network to categorize time–frequency plots of radar signals, designed to handle an unspecified number of radar signals. This technique delivered impressive Mean Intersection over Union (MioU) and Mean Pixel Accuracy (MPA) results for the modulation classification of both single and multiple radar signals in environments with high noise levels [12]. Nonetheless, semantic segmentation networks, which depend on pixel-level identification, often encounter challenges such as high computational demands, lengthy training periods, and substantial parameter sizes. Wang et al. merged the ALOHA algorithm with a binary tree algorithm to create a machine learning-based hybrid anti-collision algorithm for analyzing mixed RF signals. This strategy successfully separates signals under both positive definite and underdetermined conditions, effectively estimating the number of RF signals and showcasing strong recognition capabilities and robustness [13]. These techniques underscore diverse strategies for addressing the challenges in recognizing modulation types in mixed signal scenarios, each with distinct advantages and disadvantages. Further research is essential to develop robust and efficient methods for automatic modulation recognition in the complex and fluctuating conditions of radio frequency environments.

Current techniques for recognizing multiple signal modulations, which expand on the recognition of individual modulation types, have advanced considerably. Nevertheless, significant interference and overlap among signals present challenges in feature extraction for neural networks, necessitating more advanced methods to effectively retain information. Moreover, neural networks for signal classification often utilize multiple classifiers. For example, reference [14] describes a CNN-based method for automatically recognizing mixed-signal modulations. However, a limitation of this method is the rapid increase in classification categories as the number of modulation types increases.

Inspired by fine-grained image classification networks and methods for fine-grained signal analysis [15,16,17,18], this paper introduces a recognition model designed to achieve a detailed feature analysis of signals. The objective is to develop a high-precision recognition model that can accurately classify mixed signals across various modulation schemes, providing valuable insights for subsequent digital communication system processes. The model utilizes a ResNet framework as its core architecture to extract essential information. It incorporates a Geometric Coordinate Attention Fusion module, which applies principles of information geometry to enhance pooling methods and bolster attention mechanisms. This strategy aims to thoroughly analyze feature information of samples, integrate features across different scales and modalities, and employ low-rank bilinear pooling to capture intricate signal details by amplifying differences between signal features. This enhancement boosts modulation type recognition and reduces the need for signal preprocessing, ultimately improving classification performance. To mitigate the impact of varying signal quantities on the network’s output parameters, a multi-label classification approach is adopted for signal categorization. This method significantly reduces output dimensions relative to traditional multi-class classification techniques. In the experiments, a dataset was constructed with signals from five different modulation types, including mixed signals, under various signal-to-noise ratios to validate the effectiveness of the proposed module in enhancing classification capabilities. Additionally, experiments with a robustness dataset demonstrated the model’s stability and general applicability across diverse mixed signal conditions. These research findings provide new ideas and methods for the development of automatic modulation recognition technology, with the potential to enhance the performance and reliability of wireless communication systems in practical applications. Future research will continue to focus on optimizing model structures, improving recognition accuracy, and exploring more efficient signal feature extraction and classification methods to address increasingly complex and dynamic communication environments.

The main innovations of this paper are as follows:(1)The proposed Geometry Coordinate Attention Fusion Module is applied in neural networks. This module defines pooling operations based on information geometry theory to design an effective spatial dimension feature enhancement mechanism. This method provides corresponding time–frequency weights for features, resulting in more expressive feature representations.(2)The low-rank bilinear pooling module is introduced to achieve cross-layer interaction between fused features and backbone features. This module obtains fine-grained feature representations of signals by mapping and aggregating between different features.

The remainder of this article is as follows. In Section 2, we introduce the signal reception model, the derivation of the information geometry model, and related work, including the backbone neural networks we used. In Section 3, we present our proposed model framework and its underlying principles. In Section 4, we evaluate various performance metrics of the model through experiments and conduct a comparative analysis. In Section 5, we summarize the strengths and weaknesses of the proposed model and provide an outlook on future development directions.

## 2. Related Works

### 2.1. Signal Data

In a complex electromagnetic environment, there typically exists one or more sources that transmit and receive signals. When there are M signal transmitters and the signals are captured by a single-channel sensor, the mathematical expression of the signal can be described as follows:(1)$$x\left(t\right)=A\sum_{i=1}^{M}s_{i}\left(t\right)+n\left(t\right)$$
where $x\left(t\right)$ represents the signal received by the single-channel sensor, A denotes the signal receiving mixing coefficient matrix, $s_{i}\left(t\right)$ is the *i*-th independent signal source, M is the total number of independent signal sources that the sensor can receive, and *n*(*t*) represents additive noise.

In complex electromagnetic environments, the signals received often exhibit poor sparsity and significant overlap, whether in the time or frequency domains. Distinguishing between various signal types in these domains remains a considerable challenge. To address this, our study converts the signals into the time–frequency domain (TF), where they are represented as two-dimensional images, evolved from their original one-dimensional forms. This conversion notably enhances signal sparsity and magnifies the differences among distinct signals. We employ the Short-Time Fourier Transform (STFT) as the core method and investigate the energy plots it produces. The mathematical formula for the STFT is presented below:(2)$$\mathrm{STFT}\left(t,f\right)=\int_{-\infty }^{\infty }\left[x\left(u\right)h\left(u-t\right)\right]e^{-2j\pi fu}du$$
where $x\left(t\right)$ is the received signal, $h\left(t\right)$ is the window function of STFT, and the window function used in this article is the Hamming window.

The energy plots in the time–frequency domain are generated based on time–frequency representations. They possess the characteristic of being less affected by cross-interference, and they exhibit better separability in signal time–frequency domain.

The mathematical expression of the energy diagram is
(3)$${\left(\mathrm{STFT}\left(t,f\right)\right)}^{2}={\left(\int_{-\infty }^{\infty }\left[x\left(u\right)h\left(u-t\right)\right]e^{-2j\pi fu}du\right)}^{2}$$

The mathematical expression of the energy diagram after the signal is transformed is
(4)$$X\left(t,f\right)=A\sum_{i=1}^{M}S_{i}\left(t,f\right)+N\left(t,f\right)$$
where $X\left(t,f\right)$ is the received signal of $x\left(t\right)$ after time–frequency transformation, $S_{i}\left(t,f\right)$ is the transmitted signal of $s_{i}\left(t\right)$ after time–frequency transformation, and $N\left(t,f\right)$ is the noise in the time–frequency domain.

### 2.2. Subsection Statistical Manifolds and the Geometric Structures

In the pooling layers of neural network models, traditional average pooling and max pooling operations extract features by focusing solely on the average or maximum values within specific regions. A more sophisticated approach involves performing separate average pooling and max pooling operations on the signals while sharing weights [19], but this method still results in information loss. This paper proposes a novel approach based on information geometry theory to preserve more feature information during pooling operations. The goal is to enhance the separability between sample features and improve the model’s recognition accuracy.

#### 2.2.1. Fisher Information

Based on information geometry theory, under different principles of invariance, statistical regularity can be endowed with different Riemannian metrics, representing distinct geometric structures. The Fisher information matrix, due to its statistical and geometric properties, serves as a cornerstone in the theory and applications of information geometry, often used for constructing Riemannian metrics on statistical manifolds.

Without loss of generality, consider the density function of an exponential family, $p_{\theta }(x),\theta \in \Theta$, and
(5)$$\int_{-\infty }^{+\infty }p_{\theta }\left(x\right)dx=1,\theta \in \Theta$$
where θ is the parameter space of the probability density function of the exponential distribution family.

In fact, there are many ways to define the Riemannian metric on the parameter space Θ. For different problem scenarios, it is necessary to select a concise and effective metric among many Riemannian metrics. Without losing generality, the algorithm in this chapter selects the Fisher information metric as the analysis object.

Assume the following:(6)$$\partial_{i}=\frac{\partial }{\partial \theta_{i}}$$
then
(7)$$\int_{-\infty }^{+\infty }\partial_{i}\partial_{j}p_{\theta }\left(x\right)dx=0,\theta \in \Theta$$

Therefore, the Fisher information metric can be expressed as
(8)$$g_{ij}=\int_{-\infty }^{+\infty }p_{\theta }\left(x\right)\partial_{i}\partial_{j}p_{\theta }\left(x\right)dx$$

Assume $u,v\in T_{\theta }$, where $T_{\theta }=T_{\theta }(\Theta )$ is the tangent space at point θ∈Θ,
(9)$$u=u^{i}\partial_{i},\begin{array}{cc} & v=v^{i}\partial_{i} \end{array}$$

Therefore,
(10)$$g\left(u,v\right)=g_{ij}u^{i}v^{j}$$

#### 2.2.2. Gaussian Statistical Manifold

The Gaussian distribution has extremely broad practical applications. In radar engineering, the probability distributions of many random variables can be approximated by the normal distribution. It holds a significant position in information geometry and is one of the core topics in statistical geometric analysis.

Consider the following density function of the Gaussian distribution family:(11)$$p\left(x;\mu ,\sigma \right)=\frac{1}{\sigma \sqrt{2\pi }}\exp\left(-\frac{{\left(x-\mu \right)}^{2}}{2\sigma^{2}}\right)$$
where *μ* and *σ* are the mean and standard deviation of the Gaussian distribution, respectively. Taking the logarithm of (10), we can obtain
(12)$$\log p\left(x;\mu ,\sigma \right)=\frac{\theta_{1}^{2}}{4\theta_{2}}-\frac{1}{2}\log\left(-\frac{\pi }{\theta_{2}}\right)+\theta_{1}x+\theta_{2}x^{2}$$
where $\left(\theta_{1},\theta_{2})=(\mu /\sigma ,-1/2\sigma \right)$, let
(13)$$\phi \left(\theta_{1},\theta_{2}\right)=-\frac{\theta_{1}^{2}}{4\theta_{2}}+\frac{1}{2}\log\left(-\frac{\pi }{\theta_{2}}\right)$$

Therefore,
(14)$$\log p\left(x;\mu ,\sigma \right)=\theta_{1}x+\theta_{2}x^{2}-\phi \left(\theta_{1},\theta_{2}\right)$$

Find the partial derivatives of both ends of Equation (13) with respect to $\left(\theta_{1},\theta_{2}\right)$, and we obtain
(15)$$\partial_{i}\partial_{j}\log p\left(x;\mu ,\sigma \right)=-\partial_{i}\partial_{j}\phi \left(\theta_{1},\theta_{2}\right)$$

Therefore,
(16)$$\int_{-\infty }^{+\infty }p\left(x;\mu ,\sigma \right)\partial_{i}\partial_{j}\log p\left(x;\mu ,\sigma \right)=-\partial_{i}\partial_{j}\phi \left(\theta_{1},\theta_{2}\right)$$

And the Fisher information is
(17)$$g_{ij}=-\partial_{i}\partial_{j}\phi \left(\theta_{1},\theta_{2}\right)$$
or
(18)$$\left[g_{ij}\right]=\left[\begin{array}{cc} \sigma & 2\mu \sigma \\ 2\mu \sigma & 4\mu^{3}\sigma +2\mu \sigma^{2} \end{array}\right]$$

Based on Equation (18), this paper modifies the pooling operation within the Coordinate Attention (CA) mechanism and proposes a new attention mechanism called Geometry Coordinate Attention (GCA). The GCA aims to enhance information retention in sample features.

### 2.3. ResNet

In deep learning algorithms, particularly for deep convolutional neural networks (CNNs), increasing the depth of the network can enhance its learning capacity to a certain extent. However, as the network depth increases further, a phenomenon known as the vanishing or exploding gradient problem occurs during backpropagation, where gradients become extremely small or large when propagated back to shallow layers. This makes it challenging to update parameters effectively, resulting in model degradation and reduced classification performance. To address this issue, He et al. introduced residual networks (ResNet) in 2016, which is based on residual learning [20]. The main structure in ResNet is the residual building unit (RBU), consisting of a non-linear layer followed by an identity shortcut connection. The use of identity shortcut connections helps alleviate the difficulty of parameter optimization caused by increasing model depth. Parameter updates not only propagate layer by layer but can also be directly passed through the identity shortcut connections to shallower layers in ResNet, thereby reducing the training difficulty of the network effectively.

## 3. Geometry Coordinate Attention and Low-Rank Bilinear Pooling Network

The structure of the geometric coordinated attention and low-rank bilinear pooling network constructed in this article is shown in Figure 1. This paper uses Resnet-50 as the backbone network, adds geometric coordination attention to further improve the feature learning ability of the convolutional network, and then introduces a low-rank bilinear pooling module to achieve fine-grained analysis of signal sample features. Finally, multi-label classification is used to replace the multi-class output sample prediction values. In this section, we first introduce the principle of the geometric coordination attention module, and then introduce the multi-modal feature fine-grained fusion method of the low-rank bilinear pooling module.

### 3.1. Geometric Coordinated Attention Module

In order to effectively retain multi-level feature information, this paper designs a weighted fusion module of multi-scale features. Extract low-, medium-, and high-level features from the backbone network, so that features of different scales are optimized and fused in the time–frequency domain. The design structure of this module is shown in Figure 2.

First, the low-, medium-, and high-level features x_1_, x_2_, and x_3_ of the received signal are extracted from different layers of the backbone network. Feed the features individually into GCA, where H, W, and C represent the height dimension, width dimension, and channel dimension of the feature, respectively. Since the input feature of the neural network is the time–frequency feature plot of the signal, the width dimension and height dimension of the feature can be regarded as the time and frequency dimensions of the feature. Then, in the GCA, the time–frequency dimension weight information of multi-scale modal features is extracted and weighted, and finally, the multi-scale modal features are fused.

In GCA, the pooling operation P(x) is defined according to Equation (18) as
(19)$$P\left(x\right)=v\left[g_{ij}\right]v^{T}=v\left[\begin{array}{cc} \sigma & 2\mu \sigma \\ 2\mu \sigma & 4\mu^{3}\sigma +2\mu \sigma^{2} \end{array}\right]v^{T}$$
where v=[μ,σ]; μ and σ are the mean and variance of the input features in a certain dimension. The pooling operation method defined by Equation (19) comprehensively considers the statistical and geometric characteristics of features, and can extract more feature information than the traditional average pooling operation and maximum pooling operation.

The input features encode the dimensions along the horizontal and vertical axes through two pooling operations to generate a pair of dimension-aware feature sets. This transformation can help locate the features in spatial locations, and correlate features in time and frequency. The mathematical expression of vertical pooling coding (H Pool) is as follows:(20)$$y^{H}(h,1,C)=\sum_{0\leq k\leq W}P\left(x_{C}\left(h,k,C\right)\right)$$
where $x_{C}$ is the feature of the received signal feature in each channel dimension of the neural network, h is the minimum granularity of the height dimension, c is the minimum granularity of the channel dimension, and $y^{H}\in R^{H\times 1\times C}$ is the height direction feature of the sample in the channel dimension and is also a frequency characteristic. The mathematical expression of horizontal pooling coding (W Pool) is as follows:(21)$$y^{W}(1,w,C)=\sum_{0\leq k\leq H}P\left(x_{C}\left(k,w,C\right)\right)$$
where *w* is the minimum granularity of the width dimension and *y^W^* ∈ **R**^1×*W*×*C*^ is the width direction feature of the sample in the channel dimension and also the time feature.

Time and frequency features are concatenated in the channel dimension, and the convolution function with a convolution kernel size of (1,1) is used to obtain the time–frequency joint intermediate state features. Then, the batch normalization and a non-linear activation function are applied to reduce differences among the joint features.
(22)$$y^{H+W}=\mathrm{concat}(y^{H},y^{W})$$
(23)zH+W=ReLu(BN(conv(yH+W))),zH+W∈R(H+W)×1×Cr
where *r* is used to control the GCA parameter amount.

Then, the joint intermediate features along the channel dimension are reshaped into two features: $z_{r}^{H}\in R^{H\times 1\times (C/r)}$ and $z_{r}^{W}\in R^{1\times W\times (C/r)}$. Over-sampling operations using convolutional functions with kernel size (1,1) are applied on each feature to restore the parameter dimensions equal to the input features. Finally, a non-linear activation function is applied to each feature to obtain weights or coefficients along the time and frequency dimensions.
(24)$$z^{H}=\mathrm{sigmoid}\left(\mathrm{conv}\left(z_{r}^{H}\right)\right),z^{H}\in R^{H\times 1\times C}$$
(25)$$z^{W}=\mathrm{sigmoid}\left(\mathrm{conv}\left(z_{r}^{W}\right)\right),z^{W}\in R^{1\times W\times C}$$
where sigmoid(·) is the sigmoid function operation, which performs normalization and activation.

Finally, multiply the input features of different scales by their corresponding feature weight coefficients along the dimensions to obtain enhanced weighted features along the time and frequency dimensions. Additionally, normalize the scaled weighted features to achieve feature fusion with unified scale parameters.
(26)$$x_{i}^{H+W}=x_{i}\times z_{i}^{H}\times z_{i}^{W},i=1,2,3$$
where $x_{i}^{H+W}$ is the weighted feature obtained by multiplying the corresponding input feature with its weight coefficient.
(27)$$z=x_{1}^{H+W}⊕x_{2}^{H+W}⊕x_{3}^{H+W}$$
where the ⊕ operation is element-wise addition. The variable $z\in R^{7\times 7\times 2048}$ is the final output, which is a multi-scale weighted fusion feature.

### 3.2. Low-Rank Bilinear Pooling Module

This paper leverages a method involving the outer product of extracted neural network features from two modalities to perform weighted fusion of features, obtaining bilinear characteristics of the signal to enhance the network’s ability to analyze fine-grained features and consequently improve its capability for signal modulation recognition. Addressing the issue of parameter explosion resulting from bilinear pooling operations, the paper employs low-rank bilinear pooling to optimize the operation and reduce parameter count. Using Hadamard product in place of the outer product to interact with different features, the corresponding relationships are captured. This approach significantly reduces parameter count at the expense of certain computational complexity. In the final fully connected layer, a staged reduction in parameters is employed alongside introducing dropout to mitigate the risk of overfitting.

The bilinear pooling operation is shown in Figure 3, and its mathematical expression is
(28)$$f=\sum_{j=0}^{c_{x}}\sum_{k=0}^{c_{z}}wz_{j}x_{k}+\beta =x^{T}Wz+\beta$$
where x and z are two features that are mutually mapped. x is the backbone feature obtained from input samples through the backbone architecture, while z is the fused features output by the Geometric Coordinate Attention module from the high-, middle-, and low-level features of the samples. $C_{x}$ and $C_{z}$ are the channel dimensions of the two features, w is the mapping between the channel dimensions of the two features, W is the weight matrix, β is the bias, and f represents the output features.

Perform matrix decomposition on the weight matrix as follows:(29)$$W=UV^{T}$$
where U and V are low-rank decomposition matrices of W and m is the joint embedding dimension. Equation (28) can be further expressed in Hadamard product form, with a mathematical expression as follows:(30)$$F=x^{T}Wz+\beta =x^{T}UV^{T}z+\beta =U^{T}x\circ V^{T}z+\beta$$
where ∘ denotes the Hadamard product and F is the output feature after the mathematical expression transformation.

Using two low-rank matrices, U and V, to approximate W avoids the direct computation of the outer product of the two features as in the original bilinear pooling method. This reduction in computation and parameters decreases the parameter count from $C_{x}\times C_{z}$ to $m\times (C_{x}+C_{z})$. It can be observed that the parameter count is controlled jointly by the joint embedding dimension m and the channel numbers of the two input features. Therefore, while ensuring the features remain unchanged, it is possible to attempt multi-channel merging to reconstruct feature parameter scales, reduce the number of channels, and increase computational complexity to decrease parameter count, ensuring efficient network operation.

In this module, the mutual mapping between two sets of input features is used to extract joint representations of features across channels, enabling multi-modal bilinear pooling for improved neural network performance in signal modulation recognition within the same sample. Additionally, the use of low-rank matrix approximation for outer product computations addresses the issue of excessive parameter count in bilinear pooling, thereby mitigating potential inefficiencies and overfitting effects in neural network operations.

### 3.3. Loss Function and Evaluation Metrics

This paper’s model uses a multi-label classification approach for signal classification, which consists of multiple binary classifiers combined together. Sample labels are encoded using one-hot encoding. The normalization function connected to the model’s output layer is replaced with the sigmoid function instead of the Softmax function. The model employs a combination of the cross-entropy loss function and the binary cross-entropy loss (BCEloss) for multi-label binary classification. Compared to traditional multi-class algorithms, this method effectively reduces label dimensions and optimally utilizes the feature space of output predictions. The BCEloss not only considers the situation when the prediction value is 1 but also incorporates the impact of the prediction value being 0 on the loss. The expression for the BCEloss is as follows:(31)$$\mathrm{BCEloss}(y_{\mathrm{pred}},y_{\mathrm{true}})=-\frac{1}{p}\sum_{j=1}^{p}\sum_{i=1}^{q}[y_{\mathrm{true}}^{i,j}\cdot \log y_{\mathrm{pred}}^{i,j}+(1-y_{\mathrm{true}}^{i,j})\cdot \log(1-y_{\mathrm{pred}}^{i,j})]$$
where $y_{true}^{i,j}$ is the true value of the *i*-th label in the *j*-th sample, and $y_{pred}^{i,j}$ is the predicted value of the *i*-th label in the *j*-th sample. p is the total number of samples, which is equivalent to the batch size during batch training. q is the number of sample labels, which is equivalent to the prediction output dimension.

The evaluation indicators of the experiment are based on ref. [21], which uses the exact match ratio (EMR), hamming score (HS), and F-Measure (F1). The three indicators are used as a specific evaluation of the model’s ability to identify signal modulation types. The exact match ratio is the strictest measure of model recognition effectiveness, representing the ability of model identification to be completely accurate, but partial accuracy is also part of model evaluation. Therefore, hamming score and F-Measure are included as evaluation metrics. The hamming score refers to the accuracy feedback of the overall sample, representing the model’s perception of all labels. The F-Measure is a comprehensive measure of sample recognition accuracy, balancing precision and recall, representing the balance of the model.

The calculation formula for the exact match ratio (EMR) is as follows:(32)$$\mathrm{EMR}=\frac{1}{p}\sum_{j=1}^{p}I(y_{\mathrm{true}}^{j}==y_{\mathrm{pred}}^{j})$$
where $y_{pred}^{j}$ is the predicted value set for the *j*-th sample, $y_{true}^{j}$ is the true value set for the *j*-th sample, and I(⋅) is a function that returns 1 when the predicted value is exactly equal to the true value, otherwise it returns 0. Accuracy represents the average accuracy across all samples, where for each individual sample, it indicates the proportion of signal categories where both predicted and true values are true among the total signal categories predicted as true or true in reality.

The hamming score calculation formula is as follows:(33)$$\mathrm{Acc}=\frac{1}{p}\sum_{j=1}^{p}\frac{\sum_{i=1}^{q}y_{\mathrm{pred}}^{i.j}\cap y_{\mathrm{true}}^{i.j}}{\sum_{i=1}^{q}y_{\mathrm{pred}}^{i.j}\cup y_{\mathrm{true}}^{i.j}}$$

The F-Measure calculation formula is
(34)$$F1=\frac{1}{p}\sum_{j=1}^{p}\frac{\sum_{i=1}^{q}2\left(y_{\mathrm{pred}}^{i,j}\cap y_{\mathrm{true}}^{i,j}\right)}{\sum_{i=1}^{q}y_{\mathrm{pred}}^{i,j}+y_{\mathrm{true}}^{i,j}}$$

## 4. Simulation Experiment and Performance Analysis

### 4.1. Dataset

In this experiment, a single-antenna reception method was simulated under complex environments by adding colored noise to sample signals to better reflect real-world conditions. The dataset consists of single-signal samples or mixed-signal samples with a signal-to-noise ratio (SNR) spanning from −12 dB to 8 dB in 2 dB increments. The signal sampling frequency is 1 × 10^7^ Hz, with a frequency range from 0.5 × 10^7^ Hz to 3 × 10^7^ Hz. The resulting sample data format is a 224 × 224 RGB matrix. The mixing of signals is based on fifteen combinations as listed in Table 1, with each type of sample generated at various SNR levels resulting in 240 samples per combination, totaling 39,600 signal time–frequency energy plots. The mixing levels among different signal types are evenly distributed. The dataset is divided into training and test sets with a ratio of 5:1.

### 4.2. Experiment Environment

The experimental environment configuration for this paper is as follows: the computer system used was Ubuntu 16.04, with a TAITAN graphics card and 24 GB of memory. The programming language used was Python 3.8, the deep learning framework was PyTorch 1.7, and the optimizer was Adaptive Moment Estimation optimizer. For training, the approach involved training from scratch, with an initial learning rate set to 0.001 and adaptive decay. The training was conducted for 200 epochs, and the batch size was set to 32 based on network parameters and GPU memory size.

### 4.3. Ablation Study

To verify the reliability of the proposed modules in this paper, four ablation experiments were conducted using the same dataset and experimental setup with identical evaluation metrics. The neural network model containing both the Geometry Coordinate Attention Fusion Module and the low-rank bilinear pooling module was named GCABNet; the model containing only the Geometry Coordinate Attention Fusion Module was named GCANet; the model containing only the low-rank bilinear pooling module was named BNet; and the backbone network model was named InitNet. In the BNet model, both input variables in the bilinear pooling module use feature *x*. Additionally, a model named GCABVGG, which integrates the proposed modules into the VGG16 network structure, was compared with the VGG16 network (VGG).

The results of the ablation experiments under dynamic SNR conditions depicted in Figure 4a–c indicate that the GCA Module and the low-rank bilinear pooling module proposed in this paper led to an approximately 9% increase in EMR and a 5% increase in HS under low SNR conditions. The model’s recognition capability improved under high SNR conditions, with an EMR of over 98% for signal modulation type recognition at SNRs above 2 dB and approaching 100% at SNRs exceeding 4 dB. Specifically, the low-rank bilinear pooling module achieved an approximate 6% increase in EMR under low SNR conditions, demonstrating that bilinear pooling can yield more effective discriminative features at a finer granularity, significantly impacting the model’s recognition capability. Furthermore, the Geometric Coordinate Attention accurately obtained time–frequency dimension weights from multiple scales of sample features, effectively retaining sample feature information and assisting the model in capturing underlying discriminative features. Therefore, the proposed modules in this paper show feasibility in improving the recognition of signal modulation types.

Moreover, as shown in the experimental results in Figure 4d-f, integrating the proposed modules into the VGG network as the backbone structure resulted in a 7% improvement in EMR and approximately 9% improvement in HS under low SNR conditions, with improved balance in the F-Measure, further demonstrating the reliability of the proposed modules in this paper.

### 4.4. Attention Comparison Experiment

To verify the advantages of obtaining more effective time–frequency weight coefficients using the Geometric Coordinate Attention proposed in this paper’s GCA Fusion Module, this section conducts six comparative experiments by replacing the GCA part of the GCA Fusion Module with CA, SAM, BAM, CAM, CBAM, and SENet, while keeping all other conditions identical in the experiment environment. The contrasting performance of different attention on the dataset in terms of recognition capability and classification effectiveness is evaluated.

Specifically, CA (Coordinate Attention) analyzes the input features in multiple dimensions through multiple independent attention heads and outputs weight coefficients after normalization. SAM (Spatial Attention Module) analyzes both height and width dimensions, enhancing the model’s ability to process spatial information. CAM (Channel Attention Module) extracts channel dimension information of features, enhancing the model’s ability to learn dependencies between different channels. BAM (Bottleneck Attention Module) is a parallel combination of CAM and SAM attention mechanisms. CBAM (Convolutional Block Attention Module) is a series combination of CAM and SAM attention mechanisms. SENet (Squeeze-and-Excitation Network) enhances the response of important feature channels while suppressing the response of unimportant feature channels.

The experimental results in Table 2 demonstrate that the GCA Module used in this paper exhibits superior performance compared to classical attention. The GCA used in this paper outperforms other attention mechanisms in terms of both EMR and HS, with up to a 3% performance advantage in EMR and a 1% performance advantage in HS. Furthermore, the F-Measure results indicate that the stability of the mechanism proposed in this paper is also advantageous compared to other mechanisms. These results demonstrate that the GCA proposed in this paper maximizes the extraction of signal feature weight coefficients based on information geometry theory, enhancing the learning ability of neural networks and thereby improving model recognition capabilities. Compared to other mechanisms, the GCA considers both local granularity information in the time–frequency domain and integrates channel weights, retaining more feature information and enhancing the separability of signal features, proving the feasibility and importance of capturing time–frequency weight information. The comparison between the proposed GCA and the CA concludes that attention mechanisms based on information geometry theory can more effectively retain sample feature information, thus improving model recognition capabilities.

### 4.5. Robustness Experiment

To verify the robustness of the signal recognition model proposed in this paper, a supplementary dataset was created in this section. It includes additional mixed-signal combinations not present in the original dataset from Section 4.1. Under the same experimental conditions, this supplementary dataset was used as the test set for further experiments to validate the reliability of the signal recognition model. The signal combinations in the supplementary dataset are shown in Table 3. The robustness dataset comprises mixed signals with signal-to-noise ratios (SNRs) ranging from −12 dB to 8 dB in steps of 2 dB. Each type of sample was generated 40 times at each SNR level. In the robustness experiment, a confusion matrix was used to represent the experimental results. In this matrix, aside from signal mixture types serving as labels, the “X” label was introduced to indicate that the model recognized a signal combination not present in the corresponding dataset mixture. Additionally, the “NULL” label denotes cases where the model did not recognize any signals.

Figure 5 shows the confusion matrix obtained by randomly selecting 80 signal samples for each mixed type from the original dataset at various signal-to-noise ratios (SNRs). It reveals that the EMR for signals reaches 84.6%, with EMR exceeding 92% for single signals and nearing 80% for multiple signals. The HS is concluded to be 91.1%. It is also observed that as the number of coexisting mixed signals within a sample type increases, the model’s ability to recognize signal types weakens. This is due to the increased number of signals under the same noise conditions, causing energy dispersion in the time–frequency domain and resulting in blurred features that reduce separability and increase recognition difficulty.

In Figure 6, the EMR for signals from the supplementary dataset is only 47.2%, while the HS reaches 84.6%. The supplementary dataset contains a significantly higher number of mixed-signal samples compared to the original dataset, resulting in a noticeable decrease in the EMR due to the increased complexity of mixed samples. However, the HS approaches that of the original dataset. Considering the EMR, HS, and analysis of the confusion matrix, the model’s ability to recognize signal types in the supplementary dataset is comparable to that in the original dataset. However, there is a discernible limitation in distinguishing between different types of signal combinations, particularly affecting the recognition performance when QAM and BPSK signal types coexist. Therefore, under the same experimental conditions, introducing supplementary dataset samples with a 40:1 ratio in the training set enhances the model’s learning of different types of combinations and leads to a new supplementary confusion matrix.

In Figure 7, the EMR of the samples reaches 73%, with an HS of 90.3%. The majority of samples that did not achieve an absolute match were due to misidentifications of a single signal type, indicating a certain level of accuracy. Therefore, the model in this study demonstrates a certain capability and robustness in recognizing signal combinations outside of the training samples. After training with small samples of unknown combinations, the recognition capability improves significantly, with an approximately 17% increase in EMR and a 5.7% increase in HS, approaching the model’s accuracy on the original dataset.

## 5. Conclusions

To effectively capture and utilize the fine-grained features of mixed signals and reduce signal preprocessing, this study developed an end-to-end recognition model for analyzing the granularity of mixed signals. The model integrates the Geometric Coordinated Attention (GCA) Module to identify potential features across time and frequency domains and optimally utilize multi-scale features. Furthermore, the model employs low-rank bilinear pooling to map backbone features to optimized time–frequency features, thus enhancing the model’s capacity to discern signal sample types with greater granularity. In the ablation studies, the effectiveness of the newly introduced GCA Module and the low-rank bilinear pooling module was confirmed. By evaluating various attention mechanisms’ impacts on model performance, this study revealed that the chosen attention mechanism more effectively extracts time–frequency weights and preserves feature information, leading to enhanced recognition performance. The robustness experiments indicated the model’s capability to recognize novel combinations of signal types, but further improvements in model performance are needed. Future efforts will aim to boost the model’s recognition accuracy for different combinations of signals, while managing neural network complexity and maintaining parameter volume.

## Figures and Tables

**Figure 1 sensors-24-04530-f001:**
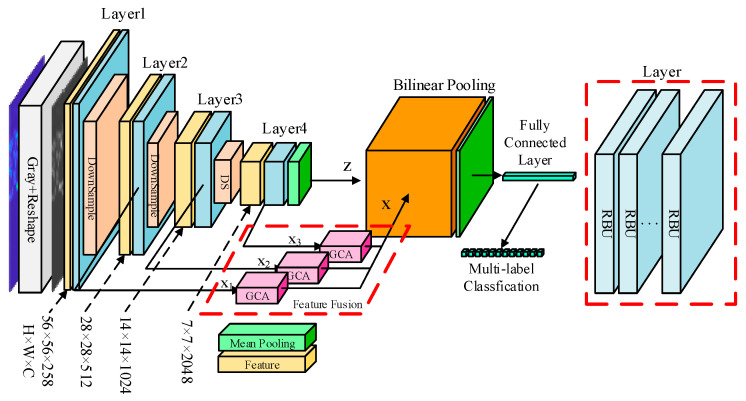
Neural network framework.

**Figure 2 sensors-24-04530-f002:**
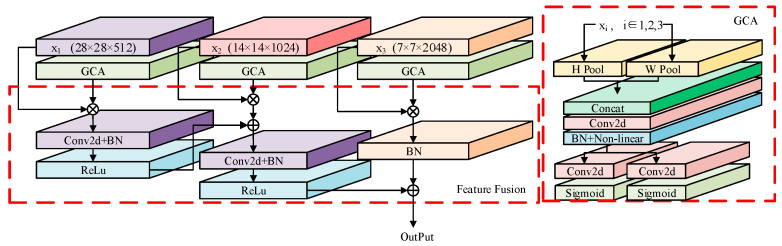
Geometry Coordinate Attention mechanism.

**Figure 3 sensors-24-04530-f003:**
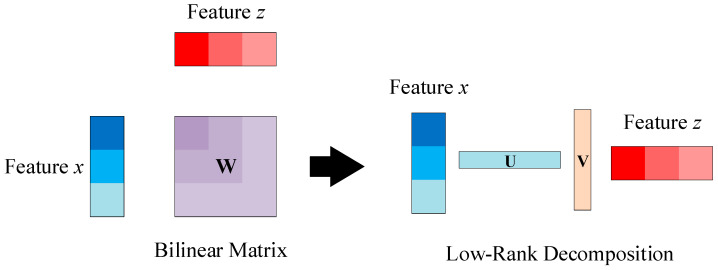
Low-rank bilinear pooling method.

**Figure 4 sensors-24-04530-f004:**
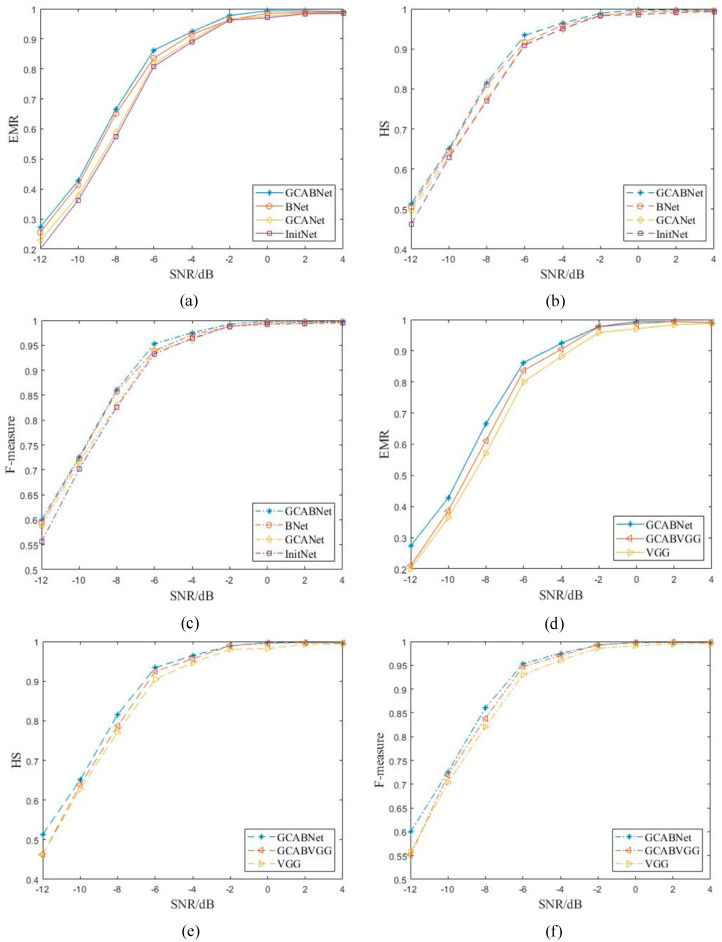
Ablation results: (**a**) EMR in Resnet-50 backbone network; (**b**) HS in Resnet-50 backbone network; (**c**) F1 in Resnet-50 backbone network; (**d**) EMR in VGG16 backbone network; (**e**) HS in VGG16 backbone network; (**f**) F1 in VGG16 backbone network.

**Figure 5 sensors-24-04530-f005:**
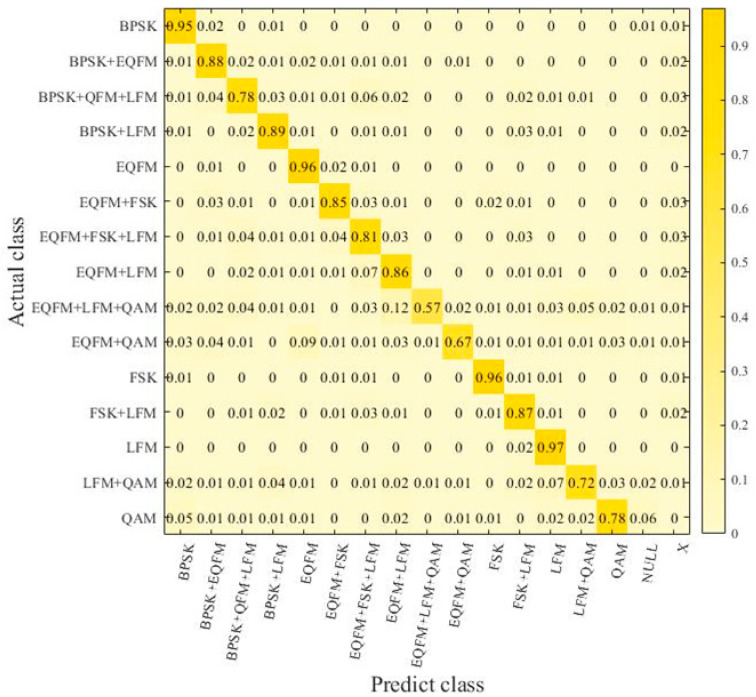
Confusion matrix in initial dataset.

**Figure 6 sensors-24-04530-f006:**
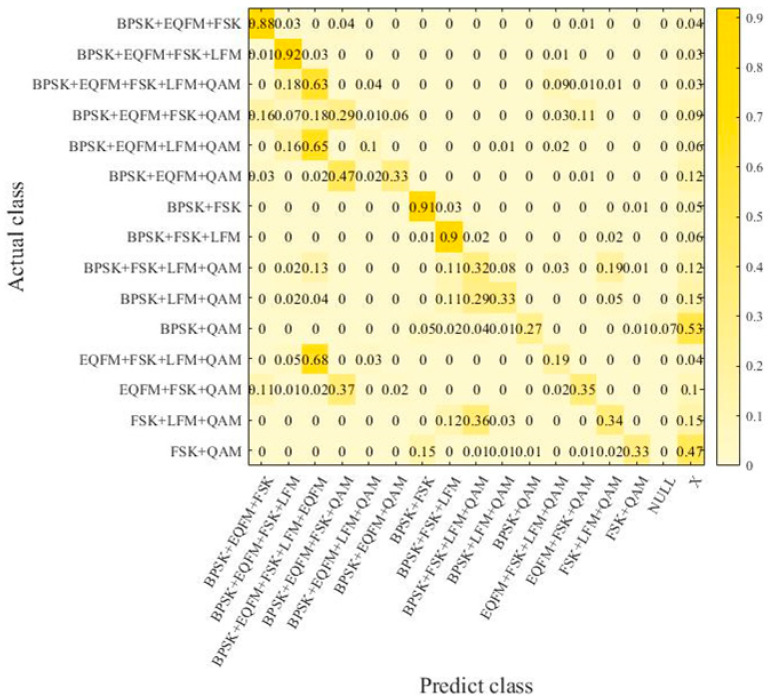
Confusion matrix in supplementary dataset.

**Figure 7 sensors-24-04530-f007:**
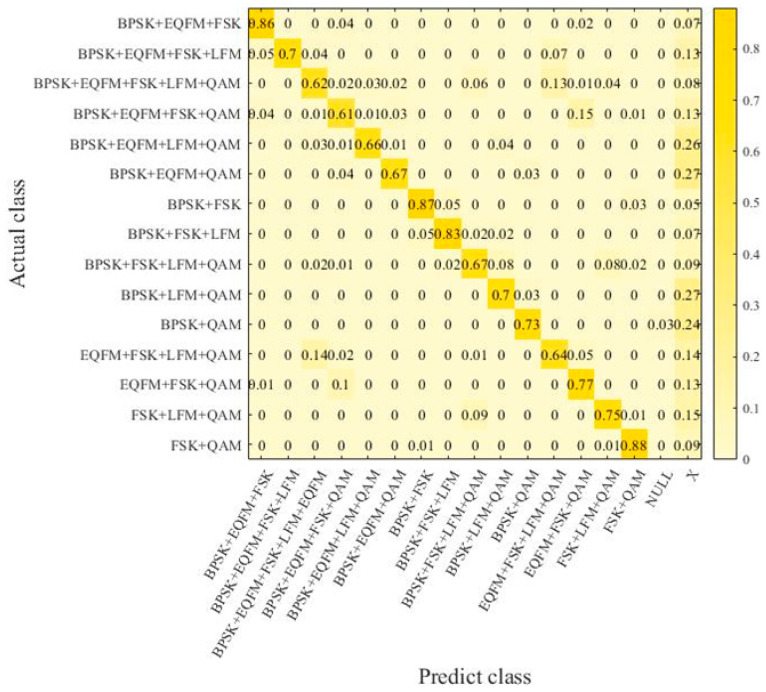
Confusion matrix in supplementary dataset with enhance training.

**Table 1 sensors-24-04530-t001:** Signal sample combination in initial dataset.

Number of Signal Modulation Types	Signal Combination
1	[LFM], [EQFM], [QAM], [FSK], [BPSK]
2	[LFM + QAM], [LFM + FSK], [LFM + BPSK], [EQFM + BPSK], [EQFM + FSK], [EQFM + QAM], [LFM + EQFM]
3	[LFM + EQFM + FSK], [LFM + EQFM + QAM], [EQFM + LFM + FSK]

**Table 2 sensors-24-04530-t002:** Performance comparison of different attention mechanisms in the model.

Attention Mechanisms	EMR	HS	F-Measure
SAM	81.313%	89.757%	92.424%
BAM	81.595%	90.024%	92.650%
CAM	81.446%	89.774%	92.364%
CBAM	82.301%	90.032%	91.831%
SENet	82.677%	90.671%	93.134%
CA	83.656%	90.851%	93.078%
GCA	84.651%	91.104%	93.143%

**Table 3 sensors-24-04530-t003:** Signal sample combination in supplementary dataset.

Number of Signal Modulation Types	Signal Combination
2	[BPSK + FSK], [BPSK + QAM], [FSK + QAM],
3	[BPSK + EQFM + FSK], [BPSK + EQFM + QAM], [BPSK + FSK + LFM], [BPSK + LFM + QAM], [EQFM + FSK + QAM], [FSK + LFM + QAM],
4	[BPSK + EQFM + FSK + LFM], [BPSK + EQFM + FSK + QAM], [BPSK + EQFM + LFM + QAM], [BPSK + FSK + LFM + QAM], [EQFM + FSK + LFM + QAM]
5	[LFM + EQFM + BPSK + FSK + QAM]

## Data Availability

Due to privacy concerns, the data supporting the findings of this study are not publicly available.

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
