# Peer review of "Fine-Grained Recognition of Mixed Signals with Geometry Coordinate Attention"

_sensors, 2024, doi:10.3390/s24144530_

Round 1
Reviewer 1 Report
Comments and Suggestions for Authors
The paper proposes an end-to-end recognition model for predicting and separating different modulated signals from the mixed signals. The model takes in the time-frequency features of the mixed signal and predicts multi-class signal types. Based on the ResNet backbone, the paper proposes to leverage Geometry Coordinate Attention (GCA) mechanisms and low-rank bilinear pooling module to fuse the multi-scale features of the input signal for an enhanced separation performance. Overall, the paper and the methodology are clear, and the results are solid. However, I have multiple questions that should be properly addressed before the manuscript could be potentially accepted.
Major concerns
1. Why STFT was used to prepare the input time-frequency signals? Will the results be different if another time-frequency analysis is used, for example, continuous wavelet transformation.
2. In GCA mechanism, why x3 is not processed in the same way as x2, i.e., passing to GCA, added with features from previous feature, and passing to Conv2d+BN? Is this specifically designed?
3. Loss function and evaluation metrics should be described at the end of methodology.
4. Simulated data is added with coloured noise while in equation (1), white Gaussian noise is added. Will this difference affect the theory of signal separation?
5. Why there are at most three types of signals mixed rather than five in the initial dataset?
6. How was BNet implemented when there is no attention mechanism? What are the x and z variables in this case?
7. The three metrics, EMR, HS, and F1, should be plotted in separate sub-figures.
8. Implementations of the attention mechanisms in table 2 should be elaborated either in results or in the supplementary materials.
9. The method is validated on the simulated dataset but not validated on any real radar datasets.
Minor concerns
1. The number of sources is equal to n in equation (4) while M in equation (1).
2. The colours of the blocks in Figure 1 are not consistent with the labels at the bottom of the figure. Feature fusion module looks like a concatenation which is not the same as described in the methodology, where x1, x2, and x3 are added in an element-wise manner. Please revise the figure to make it clear.
3. Optimisers were not described in the experimental settings.
4. It’s better to show the percentage rather than the absolute number of samples in the confusion matrix.
5. Table 3 instead of Table 1 on page 12.
6. Most sentences are clear but could be enhanced to improve the paper’s readability.
Comments on the Quality of English LanguageMost sentences are clear but could be enhanced to improve the paper’s readability, especially in the introduction.
Reviewer 2 Report
Comments and Suggestions for Authors
Obviously, the author does not have the ability to write academic papers, and I strongly recommend the following improvements:
1. This paper does not write enough innovations, and more innovations are needed.
2. This paper does not write enough references and needs to add references, which is a simple mistake.
3. The introduction of this paper has no ending, which is also a simple mistake. All authors should be held accountable for these unprofessional actions.
4. We judge that the application scenario of this paper is radar, but the paper does not emphasize the suppression of various interference methods, which is a mistake in content and will cause the method cannot be applied.
To sum up, I don't think the paper can be published.
Reviewer 3 Report
Comments and Suggestions for Authors To solve the problems of poor recognition ability and low universality caused by time-frequency overlap during signal transmission, this paper proposes a fine-grained recognition model of mixed signals with Geometry Coordinate Attention. However, there are still some issues that need further explanation and improvement in the article. 1. The comparison between the method proposed in this article and other commonly used methods is not sufficient. For example, the model in this paper can be compared with traditional feature engineering-based methods and deep learning-based methods to analyze their advantages and disadvantages in dealing with mixed signals. 2. More details about the dataset need to be described, such as how the dataset simulates mixed signals in actual communication environments, and why these signal types and signal-to-noise ratio ranges are chosen. 3. The confusion matrices in Figures 5 and 6 are not clear enough. 4. The reasons for selecting evaluation indicators in the experiment need to be further explained, as well as how they reflect the performance of the model. For example, the three evaluation indicators, Exact Match Ratio (EMR), Hamming Score (HS), and F-Measure (F1), their applicability need to be explained in evaluating model performance respectively.Author Response
Please see the attachment. Thanks.

Round 2
Reviewer 1 Report
Comments and Suggestions for Authors
The authors have addressed all my questions.
Reviewer 3 Report
Comments and Suggestions for Authors
The author has provided additional explanations and modifications to the issues raised last time. There are no other comments for the modified version.